# The ISG15-Protease USP18 Is a Pleiotropic Enhancer of HIV-1 Replication

**DOI:** 10.3390/v16040485

**Published:** 2024-03-22

**Authors:** Chaohui Lin, Edmund Osei Kuffour, Taolan Li, Christoph G. W. Gertzen, Jesko Kaiser, Tom Luedde, Renate König, Holger Gohlke, Carsten Münk

**Affiliations:** 1Clinic of Gastroenterology, Hepatology and Infectious Diseases, Medical Faculty, Heinrich Heine University Düsseldorf, 40225 Düsseldorf, Germany; linchaohui@westlake.edu.cn (C.L.); eosei@rockefeller.edu (E.O.K.); taolan.li@med.uni-duesseldorf.de (T.L.); tom.luedde@med.uni-duesseldorf.de (T.L.); 2Institute for Pharmaceutical and Medicinal Chemistry, Heinrich Heine University Düsseldorf, 40225 Düsseldorf, Germany; christoph.gertzen@hhu.de (C.G.W.G.); Jesko.Kaiser@hhu.de (J.K.); gohlke@uni-duesseldorf.de (H.G.); 3Host-Pathogen Interactions, Paul-Ehrlich-Institut, 63225 Langen, Germany; renate.koenig@pei.de; 4Institute of Bio- and Geosciences (IBG-4: Bioinformatics), Forschungszentrum Jülich GmbH, 52425 Jülich, Germany

**Keywords:** ISG15, ISGylation, USP18, p53, STING, HIV-1, SAMHD1

## Abstract

The innate immune response to viruses is formed in part by interferon (IFN)-induced restriction factors, including ISG15, p21, and SAMHD1. IFN production can be blocked by the ISG15-specific protease USP18. HIV-1 has evolved to circumvent host immune surveillance. This mechanism might involve USP18. In our recent studies, we demonstrate that HIV-1 infection induces USP18, which dramatically enhances HIV-1 replication by abrogating the antiviral function of p21. USP18 downregulates p21 by accumulating misfolded dominant negative p53, which inactivates wild-type p53 transactivation, leading to the upregulation of key enzymes involved in de novo dNTP biosynthesis pathways and inactivated SAMHD1. Despite the USP18-mediated increase in HIV-1 DNA in infected cells, it is intriguing to note that the cGAS-STING-mediated sensing of the viral DNA is abrogated. Indeed, the expression of USP18 or knockout of ISG15 inhibits the sensing of HIV-1. We demonstrate that STING is ISGylated at residues K224, K236, K289, K347, K338, and K370. The inhibition of STING K289-linked ISGylation suppresses its oligomerization and IFN induction. We propose that human USP18 is a novel factor that potentially contributes in multiple ways to HIV-1 replication.

## 1. Introduction

### 1.1. Interferon and Interferon Induction by HIV-1

The host immune response to human immunodeficiency virus 1 (HIV-1) infection likely begins with the recognition and sensing of an extracellular virion or viral molecules inside uninfected or infected cells (e.g., gastric epithelial cells, T-cells, macrophages, dendritic cells) at the portal of entry. The ensuing host–virus interactions mediate immune activation and determine the outcomes of the immune responses, the control of the virus, chronic immune inflammation, and immune pathologies, such as the death of the target cell. CD4^+^ T-cells of the mucosal epithelium potentially represent the initial targets of an infectious virion [1,2]. The structure and distinct stages of the HIV-1 replication cycle provide potential pathogen-associated molecular patterns (PAMPs) that can be recognized and sensed by the pattern recognition receptors (PRRs) in infected cells. The PAMP–PRR interaction triggers a signaling cascade within the infected cells that initiates innate intracellular antiviral effectors, which are aimed at restricting the replication and spread of the virus (Figure 1). These cell-intrinsic effectors propagate externally through the action of secreted factors such as chemokines and type I interferons (IFN-I) [3,4]. Antiviral innate immune cells can further contribute to the control of viremia and modulate the quality of the adaptive immune response to the infection [5].

Following cell entry, the HIV-1 reverse transcriptase (RT) converts the viral single-stranded (ss) RNA genome into double-stranded (ds) DNA that is ultimately integrated into the host cell genome to establish the provirus [6]. Multiple families of PRRs can sense HIV-1 viral RNA as well as the viral DNA produced during the reverse transcription of HIV [7]. HIV-1 reverse transcription intermediates such as cDNA, ssDNA, DNA/RNA hybrids, and the final dsDNA have been identified as PAMPs. They are potentially recognized by cytoplasmic DNA sensors, such as cyclic GMP–AMP synthase (cGAS), interferon gamma inducible protein 16 (IFI16), and DEAD-box helicase 41 (DDX41), which all lead to the activation of the stimulator of interferon genes (STING)-dependent antiviral immune responses [7,8,9,10,11]. In the event that HIV-1 is taken up by endosomes, the ssRNA genome of HIV-1 can be sensed by endosomal Toll-like receptors (TLRs) [12,13]. After integration, the expressed, unspliced, intron-containing HIV-1 RNA is sensed by RIG-I-like receptors (RLRs) and can induce type I interferon and proinflammatory cytokines (Figure 1) [14,15].

In HIV infection, cGAS-STING signaling is the best-studied pathway to mount an antiviral immune response against the viral DNA in the early phase of replication [9]. In contrast, some reports describe conflicting results regarding whether the reverse-transcribed viral DNA is sensed by cGAS in macrophages and in T-cells [9,16,17,18,19,20,21,22]. HIV utilizes its viral core to protect its reverse transcripts from detection by cGAS; however, the level of protection, the stability of the core, and the timing of early or accidental uncoating are under debate [23,24,25]. cGAS is recruited to the viral core in a polyglutamine-binding protein-1 (PQBP1) or non-POU (Pit-Oct-Unc) domain-containing octamer-binding protein (NONO)-dependent manner in the cytosol and nucleus, respectively, enabling cGAS to recognize the HIV-1 reverse-transcribed DNA and enhance the innate signaling in response to infection by HIV [26,27,28]. PQBP1 specifically recognizes the intact viral capsid of incoming viral particles, serving as an alarm signal to summon cGAS. The recruitment of cGAS occurs only when the core integrity is lost, suggesting that core disassembly occurs after contact with PQBP1 (Figure 1) [26].

In HIV-permissive cells, the sensing of HIV-RT products by cGAS induces the production of IFN-I [4,18,29], whereas, in HIV non-permissive resting CD4^+^ T-cells, abortive RT products likely lead to the IFI16-mediated activation of the inflammasome pathway, leading to the production of IL-1β and the initiation of pyroptosis [8,30]. The proinflammatory cytokines and IFN-I released by the HIV-1-infected CD4^+^ T-cells likely recruit other innate immune target cells, including conventional dendritic cells (cDCs), plasmacytoid dendritic cells (pDCs), monocytes, and macrophages, to the site of infection [5,8].

The host–PRR and HIV-1–PAMP interactions trigger the activation of innate immune signaling in the infected cells, which culminates in the production of type I and III IFNs, including IFN-α/β. IFN-α/β production signals back via the transmembrane IFN-α receptors 1 and 2, driving IFN-α/β production and the induction of hundreds of interferon-stimulated genes (ISGs), which help to block the replication and spread of the virus, including restriction factors such as sterile alpha motif and histidine-aspartate domain containing protein 1 (SAMHD1), apolipoprotein B mRNA editing enzyme catalytic subunit 3G (APOBEC3G), tripartite motif protein 5α (TRIM5α), and myxovirus resistance 2 (MX2) [8,31,32]. Additionally, the activated IFN-I signaling cascade induces interferon-stimulated gene 15 (ISG15) and ubiquitin-specific protease 18 (USP18) (Figure 1). ISG15 exhibits antiviral activity by conjugating with viral and cellular factors such as influenza B virus nucleo-protein and cGAS (see Section 1.2) [33,34,35]. USP18 negatively regulates type I and III IFN signaling (Figure 2) (see Section 1.3) [36,37]. USP18 is not only an ISG15-specific isopeptidase (Figure 1) but also negatively regulates nuclear factor kappa B (NF-κB) signaling (Figure 2) [37,38,39]. HIV-1 infection induces the expression of USP18 [40,41]. In HIV-1 patients, IFN-I treatment cannot induce the control of HIV-1. This failure correlates with chronic immune activation and the chronic induction of ISGs, including USP18 [40]. However, the depletion of USP18 in macrophages strongly inhibits HIV-1 replication [41]. The global effects of USP18 and ISG15 on HIV-1 replication, recognition, and sensing in innate target cells are still poorly defined. ISGylation regulates the activity of p53 (see Section 1.4) and misregulated p53 affects the antiviral protein SAMHD1 (see Section 1.5). Here, we describe our findings regarding USP18 and HIV-1 generated in the DFG Priority Program SPP 1923 Innate Sensing and Restriction of Retroviruses (see Section 2 and Section 3) [42,43]. We describe our results in the context of the current literature; however, due to the large number of topics covered here, we select and focus on the findings that are relevant to our observations.

### 1.2. ISG15

We identified ISG15 as a new regulator of the activation step of the DNA sensing pathway protein STING [44]. ISG15 is an antimicrobial protein expressed at low levels under physiological conditions [4,45]. ISG15 was first recognized as a member of the ubiquitin-like protein (Ubl) family in 1987 due to its cross-reactivity with anti-ubiquitin antibodies. It contains two ubiquitin-like β-grasp domains separated by a short linker, and both domains share approximately 30% sequence homology with ubiquitin [46,47]. The *ISG15* gene comprises two exons and encodes a 17-kDa precursor. Under physiological conditions, the ISG15 precursor can be cleaved into the mature form of the 15-kDa ISG15 peptide through the removal of eight C-terminal amino acids by USP18 and possibly by USP16, retaining a shared C-terminal amino acid motif, LRLRGG, which allows ISG15 to covalently bind to the lysine residues of the substrate [48,49,50]. In addition, ISG15 expression is induced in lipopolysaccharide (LPS)-stimulated and retinoic acid (RA)-treated cells in an IFN-I-dependent manner [51,52]. Listeria monocytogenes’ DNA-mediated ISG15 induction depends on STING, TBK-binding kinase 1 (TBK1), interferon regulatory factor 3 (IRF3), and IRF7 in a cytosolic surveillance pathway, an alternative pathway that is activated by the presence of bacterial DNA inside the cell [53]. Furthermore, the expression of both the mRNA and protein levels of ISG15 is upregulated by p53 via DNA-damaging agents such as doxorubicin, camptothecin, or ultraviolet light [54,55,56].

Unconjugated ISG15, as a free molecule, exists in two different forms: released into the serum and within the cell [57]. Although ISG15 lacks a signal peptide for secretion, it has been detected in the serum of type I IFN-treated patients and in virally infected mice [58,59]. In vitro, free ISG15 protein was found in the culture media of IFN-I-treated human lymphocytes, monocytes, neutrophils, plasmablasts, and immune and non-immune cell lines [57,60]. Only a few studies have investigated the potential ISG15 secretion pathways. It has been reported that exosomes, neutrophilic granules, secretory lysosomes, and microparticle release may be involved in the secretion of ISG15 [57,60,61]. In addition, the infection of induced pluripotent stem cell (iPSC)-derived macrophages with severe acute respiratory syndrome coronavirus 2 (SARS-CoV-2) induces ISG15 secretion in a microtubule-associated protein light chain 3 (LC3)-dependent extracellular vesicle loading and secretion pathway [62].

Lymphocyte function-associated antigen 1 (LFA-1) is the receptor for extracellular ISG15 [63]. The direct interaction of extracellular ISG15 with LFA-1 initiates the activation of the SRC family of protein tyrosine kinases, stimulating IFN-γ and interleukin-10 (IL-10) secretion in natural killer (NK) cells [63]. Additionally, ISG15, constitutively produced by IFN-I-treated monocytes and lymphocytes, can induce the release of IFN-γ from T-lymphocyte cells [64]. Human ISG15 increases the proliferation and lysis of NK cells [65]. Extracellular free ISG15 can act as an adjuvant for cytotoxic CD8^+^ T-cells, thus enhancing the magnitude and quality of CD8^+^ T-cell responses to modulate antitumor immunity [66]. ISG15, by inducing E-cadherin expression in human dendritic cells, may affect the migration of these cells [67]. With respect to the impact of ISG15 on neutrophils, ISG15 acts as a chemoattractant and an activator of neutrophils [68]. The ISG15 inside exosomes and microparticles contributes to the stimulation of macrophages to regulate the transmission of anti-HIV activity and release proinflammatory cytokines, respectively [61]. Conflicting results have been obtained regarding the effect of ISG15 on macrophage phagocytosis and the generation of nitric oxide and reactive oxygen species [69]. Free ISG15 secreted from SARS-CoV-2-infected cells correlates with the expression of inflammatory genes and cytokines and the polarization of macrophages to the M1 state [62].

The main role of free intracellular ISG15 is to interact with intracellular proteins and regulate their function. A few examples of ISG15-regulated proteins are described here. An important sensor for cytosolic viral RNA is the retinoic acid-inducible gene I (RIG-I). Leucine-rich repeat-containing protein 25 (LRRC25) can interact with ISG15-associated RIG-I and mediate the degradation of RIG-I through p62-targeted selective autophagy, leading to the inhibition of IFN-I signaling during RNA virus replication [70,71]. ISG15, linked to histone deacetylase 6 (HDAC6), promotes the autophagic clearance of ISG15 conjugates [72]. Moreover, E3 ligase neuronal precursor cell-expressed developmentally downregulated 4 (NEDD4) catalyzes the ubiquitination of the matrix protein of Ebola virus VP40, facilitating the release of virus-like particles [73]. ISG15 can bind to NEDD4 and decrease the ubiquitination of VP40 to inhibit virion egress [74]. The binding of ISG15 with hypoxia-inducible factor 1-alpha (HIF-1α) impairs HIF-1α-targeted gene expression and cancer cell proliferation. S-phase kinase-associated protein 2 (SKP2) is an E3 ligase that mediates the ubiquitination of USP18 and subsequently promotes its proteasomal degradation [75,76]. However, the noncovalent binding of free intracellular ISG15 with USP18 inhibits SKP2-mediated USP18 degradation [75,77]. These results suggest that free intracellular ISG15 is essential in maintaining the long-term stabilization of USP18. In contrast, murine USP18’s stability is independent of ISG15 [44,78].

### 1.3. USP18

USP18, a member of the ubiquitin-specific protease (UBP) family with a molecular mass of 43 kDa, was first identified in mice expressing acute myeloid leukemia 1 (AML1)-eight twenty-one (ETO) (AML1-ETO) and later confirmed in virus-infected porcine alveolar macrophages and IFN-I-treated human melanoma cell lines [79,80,81]. USP18 functions as a protease to cleave ISG15 molecules from substrate proteins through isopeptide bonds (Figure 1) and as a negative regulator of IFN-I signaling [39,49] (Figure 2). Recently, USP16 was identified as another ISG15 cross-reactive protease [50]. USP18 is highly abundant in the liver, spleen, and thymus, with low levels detected in the bone marrow, adipose tissue, and lung tissue. The high expression of this protein is observed in myeloid lineage cells, including CD169^+^ macrophages, bone-marrow-derived dendritic cells, peritoneal macrophages, and monocyte-derived macrophages. USP18 expression is also observed in splenic T- and B-cells, with high abundance in naïve, effector/memory, and natural regulatory T-cells. A high expression of USP18 is also detected in Th0, Th1, and Th17 CD4^+^ T-cells [82]. USP18’s expression levels are differentially regulated during T-cell activation, tolerance, and effector differentiation. USP18 is necessary for Th17 differentiation and the development of CD11b^+^ dendritic cells [82,83]. USP18 expression is strongly induced by type I and type III IFNs and robustly upregulated following LPS, polyinosinic: polycytidylic acid (poly I:C), or tumor necrosis factor alpha (TNF-α) stimulation [39]. In line with this, viral or bacterial infection increases the expression of USP18 in cells [84,85,86].

USP18 has also been described to remove ubiquitin from some ubiquitinated proteins (Figure 2). Many studies have indicated that USP18 functions as a deubiquitination enzyme in the regulation of signaling pathways. USP18 inhibits the NF-κB-signaling-mediated regulation of T-cell proliferation and IL-2 production by catalyzing the deubiquitination of the TGFβ-activated kinase/TAK-binding protein (TAB1/TAK1) complex, which provides evidence for USP18’s regulation of T-cell-mediated autoimmunity [82]. Further studies found that USP18, but not its protease inactive form, USP18-C64S, abolished the K63-linked polyubiquitination of TAK1, suggesting that USP18 deubiquitinates TAK1 in a protease-dependent mechanism [38]. However, USP18 blocks the K63-linked ubiquitination of the NF-κB essential modulator (NEMO) in a protease-independent manner [38]. Recently, studies have demonstrated the role of USP18 in the regulation of cancer proliferation, migration, and invasion. USP18 promotes glioblastoma cell invasion and migration by eliminating the ubiquitination of twist-related protein 1, thereby preventing its degradation [87]. In colorectal cancer, USP18 enhances the proliferation of cancer cells by modulating the ubiquitination of the zinc finger protein SNAI1 [88].

Additionally, USP18 acts as a positive regulator of reactive astrogliosis by directly interacting with SRY-box transcription factor 9 (SOX9) and removing ubiquitin from SOX9, thus stabilizing the SOX9 protein [89]. Interestingly, USP18 expression has been shown to be upregulated in esophageal squamous cell carcinoma (ESCC) patients. In this cancer, USP18 enhances the protein stability of zinc finger E-box-binding homeobox 1 (ZEB1) through the decreased ubiquitination of ZEB1, increasing the migration and invasion abilities of ESCC cells [90].

USP18’s stability depends on free intracellular ISG15, and human cells lacking ISG15 exhibit prolonged ISG expression due to the loss of USP18 stabilization by ISG15 [78] (Figure 1). USP18 is a negative regulator of IFN-I signaling, independent of its ISG15 isopeptidase activity (Figure 2). Humans and mice with USP18 deficiency develop severe interferonopathies associated with upregulated interferon signaling [91,92]. In this mechanism, the signal transducer and activator of transcription (STAT2) recruits USP18 to bind with IFN-I receptor subunit 2 (IFNAR2), resulting in competition with JAK1 for association with IFNAR2, thereby downregulating IFN signaling and IFN-stimulated gene expression [93]. The disruption of the STAT2–USP18 interaction promotes the activation of IFN signaling [93]. Similarly, USP18 binds the colony-stimulating factor 1 receptor (CSF1R) and blocks the interaction of CSF1R with the ubiquitin E3 ligase NEDD4 and the ubiquitination and degradation of CSF1R [94] (Figure 2). The enzymatic activity of USP18 is not responsible for the downregulation of CSF1R. In this system, the downregulation of CSF1R through the deletion of USP18 in myeloid cells suppressed tumor progression [94]. USP18 can also indirectly regulate the innate immune responses. USP18 upregulates innate antiviral immunity by facilitating the TRIM31-catalyzed ubiquitination of MAVS, functioning as a scaffold protein, whose effect is independent of USP18’s enzymatic activity [95]. In addition, USP18 can recruit the deubiquitinase USP20 to deconjugate the ubiquitination of STING and enhance the stability of STING and the induction of IFN-I and inflammatory cytokines during DNA virus infection [96].

Recently, a study demonstrated that USP18 is also located in the nucleus and inhibits pyroptosis in tumors (Figure 2). Nuclear USP18 cooperates with NF-κB to reduce the binding of IFN-regulated transcription factors to their corresponding DNA motifs. USP18 suppression enhanced the expression of both typical and non-canonical ISGs without interacting with IFNAR2, thereby inducing pyroptosis in cancer cells [97]. However, in muscle cells, nuclear USP18 is a key regulator of the initiation of muscle cell differentiation, independently of ISG15 and its role in the ISG response (Figure 2). This initiation of differentiation was concomitant with the reduced expression of the cell cycle gene network and an alteration in the expression of myogenic transcription factors [98].

### 1.4. P53

In our study, focusing on how USP18 enhances HIV-1 replication, we identified p53 as a key player in HIV-1 replication [99]. p53 was initially discovered as a cellular protein interacting with the large T-antigen in simian virus 40 (SV40)-infected cells and was recognized as a tumor suppressor in 1989 [100,101,102,103]. The p53 protein consists of 393 amino acids and four major functional domains, including an amino-terminal transactivation domain, a core sequence-specific DNA-binding domain, an oligomerization domain, and a regulatory domain [104]. The importance of tumor suppressor p53 is irrefutable and it is commonly referred to as the “guardian of the genome”. p53 is a complex, multifunctional, sequence-specific DNA-binding transcriptional regulator that transactivates dozens of target genes involved in cell cycle arrest, DNA repair, apoptosis, and differentiation in damaged cells [105]. Accordingly, due to somatic mutations in the TP53 gene, mutations in the p53 protein are observed in a large fraction of many different types of human cancer [106]. Mutations in TP53 can lead to the loss of its tumor suppression function and, interestingly, gains of function, which may promote tumor growth [107]. Some genetic alterations in the DNA-binding domain usually result in the inhibition of p53’s binding with DNA. Other mutations destabilize the secondary structure of p53 [108].

Studies have shown that ISGylation modulates the stability and activity of p53. ISG15 is covalently conjugated to misfolded p53, thus promoting the degradation of misfolded p53 through the 20S proteasome and maintaining functional native forms of p53 activity [109]. It was confirmed that the deletion of ISG15 induced the accumulation of both misfolded and native forms of p53 in V-Src-transformed mouse embryonic fibroblasts (MEFs) [110]. In human myeloid THP-1 cells overexpressing USP18 or lacking ISG15, we found the accumulation of misfolded, high-molecular-weight p53, which inactivated the native p53’s function and gained dominant-negative activity [99]. Notably, the ISGylation of p53 can be catalyzed by different E3 ligases. Homologous to the E6-AP carboxyl terminus (HECT) and RCC1-like domain (RLD), the domain containing E3 ubiquitin protein ligase 5 (HERC5) acts as an E3 ligase, interacting with p53 to achieve 20S proteasome-mediated degradation [109]. Interestingly, another study demonstrated that DNA-damaging agents induced the E3 ligase estrogen-responsive finger protein (EFP) to target p53 for ISGylation [111].

p53 controls the transcription of genes via two functionally specialized transactivation domains that are associated with the regulatory functions of p53, including a sequence-specific DNA-binding domain (DBD) and a sequence-independent C-terminal domain (CTD) [112]. DNA-binding assays demonstrated the high-affinity binding of the p53 DBD to the consensus response elements (RE), whereas the CTD has been shown to bind DNA without sequence specificity [113]. Mutations in DBD, such as R273H, can disrupt the binding ability of p53 by directly destabilizing protein–DNA contacts, while some other mutations, such as R175H, R249S, and R282Q, can eliminate the binding by destabilizing the structure of p53 [112]. It has been described that p53 CTD phosphorylation or acetylation regulates p53’s binding to consensus REs [114,115]. Deletions at the CTD or point mutations at K320 and K382 inhibit p53-mediated transcription in the context of DNA [115]. A well-studied example is the binding of p53 to elements in the cyclin-dependent kinase inhibitor 1A (CDKN1A/p21) promoter upon DNA damage or other stressors [116]. The p21 protein then blocks the activity of several cyclin–CDK complexes and inhibits their kinase activity [117]. Misfolded, dominant-negative p53 blocks the transcription of p21 [99].

p53 is considered a regulator of both innate and adaptive immunity, directly transactivating key regulators of immune signaling pathways. The expression of several immune response genes is activated by p53, such as ISG15, protein kinase R (PKR), interferon regulatory factor 5 (IRF5), IRF9, and Toll-like receptor 3 (TLR3), which are involved in driving IFN production [103]. The ectopic expression of p53 stimulates the expression of STING, interferon-induced protein with tetratricopeptide repeats 1 (IFIT1), and IFIT3, which are antiviral effector proteins [118]. The activation of wild-type p53 upregulates both the mRNA and protein levels of UL16 binding protein 2 (ULBP2) and then promotes NK-cell-mediated target recognition and antitumor responses [119]. In addition, p53 inhibits the transcription of cluster of differentiation 43 (CD43) in non-hematopoietic cells [120]. A lack of p53 in mice led to the decreased expression of antiviral gene responses and impaired dendritic cell activation upon avian influenza virus (AIV) infection [121]. In recent years, evidence has indicated that p53 plays a protective role against the development of various autoimmune diseases by decreasing the production of proinflammatory factors. A dominant-negative mutation of p53 in rheumatoid arthritis synovial tissue has been associated with the increased local expression of interleukin-6 (IL-6). However, native p53 effectively inhibits the IL-6 promoter [122]. Other evidence suggests that p53 expressed in T-cells acts as a suppressor of the control of autoimmunity by inducing regulatory T-cell differentiation [43,123].

### 1.5. Restriction by SAMHD1

SAMHD1 is an HIV restriction factor that blocks retroviral replication in myeloid and resting CD4^+^ T-cells (Figure 1). In the first experiments on USP18 and HIV-1, we observed that USP18 expression could prevent SAMHD1’s restriction to HIV-1. As we found that USP18 formed a complex with the E3 ubiquitin ligase recognition factor S-phase kinase associated protein 2 (SKP2) and SAMHD1, we initially assumed that the protein stability or ISGylation of SAMHD1 would affect the antiviral activity of SAMHD1 [124]. However, in later experiments, we highlighted p21 as the key regulator of SAMHD1’s activity. Unlike HIV-2, which has VPX to counteract SAMHD1, HIV-1 possesses no accessory protein for SAMHD1 degradation. The VPX of SIV/HIV-2 targets SAMHD1 for polyubiquitination and proteasomal degradation by recruiting the host cell cullin-4 (CUL4) ligase substrate receptor DDB1- and CUL4-associated factor 1, DCAF1 [125,126,127,128,129,130]. VPX is packaged into the viral particles during assembly and functions early in the infection of a new cell [131]. The reason that HIV-1 lacks an accessory protein to overcome SAMHD1’s restriction function is currently under debate [132]. SAMHD1 is a dNTPase regulating the dNTP homeostasis in cells and can lower the substrate for the HIV-1 reverse transcriptase below the level required to support viral DNA synthesis [132,133,134,135]. The provision of exogenous deoxyribonucleotides (dNs) rescues HIV-1 replication in cells with active SAMHD1 [127,134]. However, other reports provide early evidence that the dNTPase activity of SAMHD1 might not solely be responsible for the restrictive phenotype [136]. Recent data point towards the lack of influence of the global cellular dNTP levels on HIV-1 restriction, despite the importance of the catalytic core of SAMHD1 [137]. The gene editing of specific known catalytic sites within SAMHD1 proves that the catalytic pocket must be intact for antiviral restriction, even though there is a poor correlation among the cellular dNTP levels and antiviral activity [137]. Other functions of SAMHD1 impacted by these residues could contribute to HIV-1 restriction—for example, the described nuclease activity of SAMHD1 [138,139]. Another plausible explanation might be that the dNTPase activity itself is important, but it is necessary to modulate the availability of dNTPs within sub-cellular compartments at the sites of HIV replication.

SAMHD1 is widely expressed in a diverse range of human tissues [140], but it appears to only restrict HIV replication in non-dividing cells. The SAMHD1 mRNA is induced by type I IFNs in a cell-type-specific manner [141,142]; however, the expression of SAMHD1 alone appears to be insufficient to confer resistance against HIV-1 in some cell types [143,144]. SAMHD1 is post-translationally regulated by phosphorylation at residue T592 in a cell-cycle-dependent manner [145]. SAMHD1 is phosphorylated by CDK1 and CDK2 in complex with cyclin A2 in S and the G2/M-phase [143,144]. At the mitotic exit, SAMHD1 is dephosphorylated at T592 by the PP2A-B55α phosphatase complex, leading to anti-HIV-1 active SAMHD1 [145]. Moreover, even in non-cycling monocyte-derived macrophages (MDMs), the antiviral activity of SAMHD1 is actively controlled by PP2A through de-phosphorylation [145]. In MDMs, the phosphorylation of T592 of SAMHD1 and the deactivation of SAMHD1’s antiviral activity are dependent on the variant expression of cell-cycle-associated proteins, which explains the occurrence of replication despite SAMHD1 [146]. Macrophages have been proposed to exist in two states through which all cells periodically cycle, characterized by a typical G0 state with a lack of the cell cycle marker minichromosome maintenance complex 2 (MCM2), which has anti-HIV-1 active and dephosphorylated SAMHD1. The second state is described as G1-like and characterized by the expression of MCM2 and anti-HIV-1 inactive phosphorylated SAMHD1 [145,146]. Intriguingly, the IFN stimulation of MDMs leads to the specific upregulation of the PP2A-B55α subunit controlling SAMHD1 dephosphorylation and antiviral activity [145].

## 2. USP18 Expression Enhances HIV-1 Replication

Aiming to understand the role of USP18 in the replication of HIV-1 in myeloid target cells, we observed that USP18 supported the replication of HIV-1 and HIV-2 through mechanisms that depended on both its interferon regulation and protease functions [124]. We found that HIV-1 infection could induce the USP18 protein in myeloid cell lines. The ectopic expression of USP18 in undifferentiated THP-1 cells increased HIV-1 replication up to 11-fold. Similarly, in the presence of VPX and even in the absence of VPX, differentiated THP-1.USP18 cells showed significantly increased HIV-1 replication compared to vector control THP-1 cells, suggesting the loss of the SAMHD1 restriction via USP18 expression [124]. Due to SAMHD1’s activity, resting CD4^+^ T-cells or myeloid cells are highly inhibitory towards HIV-1 replication, requiring higher dNTP levels for efficient reverse transcription [132]. The SAMHD1 restriction factor is positively regulated by p21, which inhibits the cyclin and CDK complex, causing SAMHD1’s phosphorylation. In the differentiated THP-1.USP18 cells, in contrast to THP-1 wild-type control cells, the phosphorylation signal of SAMHD1 was retained. We found that USP18 induced the downregulation of the p21 protein and elevated expression levels of cyclin D1 and cyclin D2 in differentiated THP-1 cells. p21, which is also an IFN-I-inducible gene, is a key factor for the growth and differentiation of myeloid cells in vivo [43,147]. Interestingly, USP18 binds to SKP2, an interaction partner of p21 and also precipitates SAMHD1 [124]. Furthermore, the p21 protein levels were upregulated in USP18 KO cells, strongly suggesting that p21’s levels are regulated by USP18. In THP-1.USP18KO cells, HIV-1 was impaired, in line with increased p21 protein levels. USP18 overexpression also reduced the p21 protein levels in THP-1.SAMHD1 KO cells. In all tested cells, the depletion of USP18 stabilized the p21 expression and consequently abrogated HIV-1’s replication [148,149].

To understand how USP18 promotes HIV-1 replication, we analyzed the level of reverse transcription during infection and found that USP18 increased both early and late reverse transcription products. Thus, we conclude that USP18 increases HIV-1 replication at the reverse transcription step. p21 functions in a cell-cycle-dependent manner, such that, in resting myeloid cells, p21 downregulates key enzymes such as the RNR2 subunit of the ribonucleotide reductase, which is involved in the de novo dNTP biosynthesis pathway [147,150]. Furthermore, in resting myeloid cells, the deoxynucleoside triphosphohydrolase activity of SAMHD1 is enhanced by dephosphorylation at T592 through the p21-dependent elimination of cyclinA/CDK1/2. In strong contrast, the expression of USP18 in differentiated and undifferentiated THP-1 cells causes the upregulation of enzymes involved in de novo dNTP biosynthesis (shown for RNR2 and TYMS) and the increased phosphorylation of SAMHD1. Together, the increased de novo synthesis of dNTPs and reduced dNTPase activity of SAMHD1 explain how USP18 enhances HIV-1’s reverse transcription.

We tested genes that mediated p21 expression, which could have been regulated by USP18. To this end, we explored several gene candidates, including p53, which is known to regulate p21. Surprisingly, despite the highly abundant p53 mRNA and protein expression in USP18-expressing myeloid THP-1 cells, the mRNA and protein levels of p21 remained low [99]. This observation motivated us to characterize the nature of the expressed p53 protein. A biochemical assay suggested that this abundant p53 protein was non-functional, high-molecular-weight, dominant-negative p53, which had the ability to counteract the wild-type p53 induction of p21 expression [99]. This misfolded dominant-negative p53 required an ISG15 modification for degradation; thus, in the presence of USP18, the dominant-negative p53 accumulated, forming prion-like aggregates that reduced the p21 expression. Similarly, the depletion of ISG15 in THP-1.ISG15 KO cells caused the accumulation of misfolded dominant-negative p53, which strongly supported HIV-1 replication. The HIV-1-enhancing effect of USP18 was dependent on its protease activity, and a USP18 variant with the catalytic site mutated to alanine (C64A) did not accumulate misfolded dominant-negative p53 [99]. Thus, we found that p53 was modified by ISG15 and that USP18 eliminated the ISGylation of p53, supporting the notion that misfolded p53 accumulates due to de-ISGylation. The accumulation of high-molecular-weight p53 with amyloid fibrils in USP18-expressing or ISG15 KO cells is reminiscent of the phenotype exhibited by the hotspot R273H mutant of p53, which has a strong tendency to form aggregates [151]. Interestingly, we found that HIV-1 infection could transiently induce p53 in myeloid cells. Finally, the overexpression of gain-of-function mutants of p53 (e.g., R273H) but not wild-type p53 enhanced the cell permissivity for HIV-1, supporting the notion that the expression of misfolded p53 enhances HIV-1 replication [99].

## 3. ISG15 Is a Regulator of HIV-1 DNA Sensing

The cGAS–STING–TBK1-IRF3 signaling axis is recognized as an essential mechanism for the DNA’s innate immune response to viruses. In this pathway, it was demonstrated that ISGylation is involved in regulating the activity of the DNA sensor cGAS, which enhances cGAS’ oligomerization and activity after Herpes simplex virus 1 (HSV-1), also known as Human alphaherpesvirus 1 (HHV-1), infection [33,35]. IRF3, the key factor for interferon responses, can also be modified by ISG15, e.g., upon Sendai virus infection. The ISGylation of IRF3 enhances the cellular antiviral responses by inhibiting the ubiquitylation-mediated proteasomal degradation of IRF3 [152]. Furthermore, ISG15 can promote antiviral immune responses through the ISGylation of the RNA sensor pathway proteins RIG-I and mitochondrial antiviral signaling protein (MAVS), as well as signal transducer and activator of transcription 1 (STAT1) [153,154,155]. Interestingly, SARS-CoV-2 counteracts the ISGylation of IRF3 and MAVS by using a viral papain-like protease that, as with USP18, cleaves ISG15 from IRF3 and MAVS to impair the IFN-I response and promote viral spread [153,156].

As we found that USP18 significantly enhanced HIV-1 replication, we expected that this would correlate with higher DNA sensing and interferon production. However, our study showed very low background levels of interferons in HIV-infected USP18 cells [42]. To address the functional impact of ISG15 on HIV-1 sensing, ISG15 KO THP-1 cells were analyzed. The ISG15 deficiency blocked interferon production following HIV-1 infection in differentiated and non-differentiated THP-1 cells. Similarly, no interferon production was detected if ISG15 KO cells were infected with the poxvirus Modified Vaccinia Virus Ankara or transfected with herring sperm DNA, while both agents triggered a strong interferon response in wild-type cells. In addition, we found that a potent pharmacologic STING agonist failed to trigger the cGAS–STING pathway in ISG15 KO cells. Thus, our findings showed that ISG15 modulated the sensing of HIV-1 and cytoplasmic DNA in a STING-dependent manner [44].

The binding of HIV-1 DNA by cGAS produces a cyclic guanosine monophosphate–adenosine monophosphate (cGAMP) as the second messenger that binds to and activates the adaptor protein STING [9,157]. The activation of STING leads to the activation of the TBK1–IRF3-signaling-mediated induction of IFN-I and antiviral interferon-stimulated genes [7,9,158,159]. Various post-translational modifications (PTMs) of STING that modulate the activation of STING-dependent signaling, such as ubiquitination, phosphorylation, SUMOylation, and acetylation, have been reported [160,161]. However, few studies have described the role of PTMs in STING in the context of HIV-1 replication. The ubiquitination of STING may be directly inhibited by HIV-1’s proteins p6 and Vif to reduce the antiviral immune response [162,163].

In our study, we found that the ISG15 modification of STING plays a critical role in HIV-1 DNA sensing [44]. STING undergoes ISGylation at lysines K224, K236, K289, K338, K347, and K370 in response to cytosolic DNA stimulation or HIV-1 replication, which promotes STING oligomerization (Figure 3A) [44]. Furthermore, we found that K289-linked ISGylation on STING was important for STING-triggered signaling activation in reconstituted STING THP-1 cells, as well as in a STING knock-in model of human-induced pluripotent stem cells (iPSCs) [44]. Molecular dynamics simulations revealed that the ISGylation of STING at K289 structurally rigidified both the orthosteric binding site and the region involved in rotation during activation. Furthermore, ISGylation might stabilize STING’s oligomerization by facilitating additional interactions between dimers (Figure 3B,C and Figure 4) [44]. In addition to its role in the regulation of cytoplasmic DNA challenges, ISGylation is required for the constitutive activity of STING-associated vasculopathy with onset in infancy (SAVI), which leads to STING activation independently of an interaction with its ligand, cGAMP [164]. The loss of ISGylation of STING–SAVI significantly reduces its oligomerization and activity [44].

## 4. Outlook

A high basal expression level of USP18 in some HIV-1 target cells or the virus-triggered upregulation of USP18 may enable the virus to bypass the intrinsic antiviral responses. In the early replication phase of HIV-1, USP18 expression is beneficial for HIV-1, and it renders the cell more permissive by reducing the antiviral activity of SAMHD1, increases the dNTP levels, blocks the IFN receptor signaling, and prevents the activation of STING (Figure 5). Whether USP18 also enhances the later stages of HIV-1 replication has not been studied. Our findings were obtained in myeloid cells; whether USP18 expression has a virus-enhancing effect in T cells is unknown [165]. However, it has been described that ISG15 expression prevents the late stages of HIV-1 replication. ISGylation disrupts the association between the ATPase vacuolar protein sorting-associated protein 4 (VPS4) and the retrovirus budding complex in the presence of the ISGylation of the ESCRT-III protein charged multivesicular body protein 5 (CHMP5) [166]. Another study demonstrated that ISG15 expression specifically inhibited the assembly and release of HIV-1 virions through the ISGylation of tumor susceptibility gene 101 protein (TSG101), leading to the disruption of the interaction of Gag–TSG101 [167]. Moreover, the ISG15 E3 ligase HERC5 restricted the replication of HIV-1 by blocking HIV-1 Gag particle production, which was correlated with the post-translational modification of Gag with ISG15, suggesting that there may be different molecular mechanisms by which ISGylation inhibits HIV-1 [168].

In conclusion, although the inducible or constitutive physiological expression of USP18 in vivo is crucial in maintaining a homeostatic balance between interferon expression and the prevention of interferonopathy, viruses such as HIV-1 may hijack this key protein to escape recognition by the innate immune system. The emerging diverse actions of USP18, as the main negative regulator of the interferon system, the main de-ISGylase, and a transcription regulator, suggest that this essential human host factor may not be a suitable pharmacological target for HIV-1 infection, and the more specific inhibition of its interaction partners may represent a strategy to target virus replication.

## Figures and Tables

**Figure 1 viruses-16-00485-f001:**
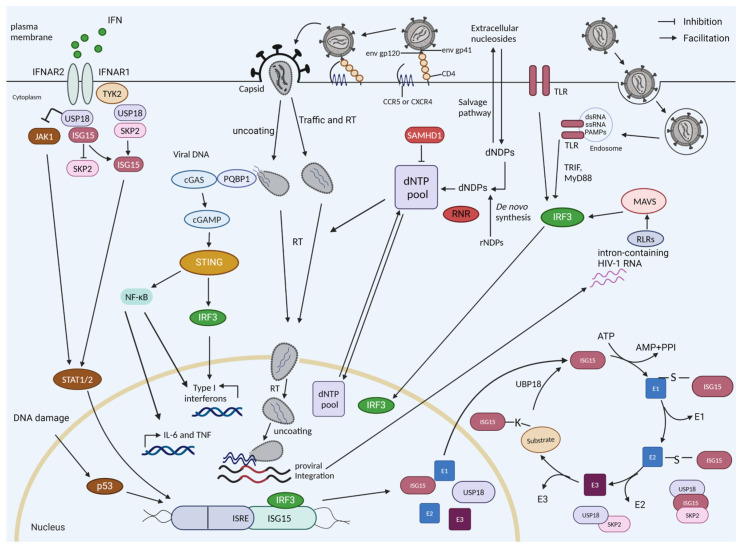
HIV sensing, interferon induction, and ISGylation in myeloid cells. The reverse-transcribed HIV-1 DNA is sensed by cGAS, which is recruited by PQBP1, which initially recognizes the viral core. cGAS-generated cyclic guanosine monophosphate–adenosine monophosphate (cGAMP) stimulates STING, which triggers a signaling cascade of TBK1-IRF3 to drive the expression of interferon-stimulated genes (ISGs) and interferons. During infection, the cell can detect the HIV RNA if the virus enters endosomes via Toll-like receptors (TLRs). The intron-containing RNA that is expressed by proviruses can be detected by RIG-I-like receptors (retinoic acid-inducible gene-I-like receptors (RLRs)). HIV-1 replication is restricted in resting cells by SAMHD1, a dNTPase. The dNTP level will also be regulated by de novo synthesis (e.g., by RNR). ISG15 and USP18 are ISGs. USP18 is a negative regulator of the interferon pathway; it competes with JAK1 binding at the interferon receptor, thereby blocking its signaling. USP18 is the main de-ISGylase that removes covalently bound ISG15 from ISGylated proteins. ISGylation is a process involving E1, E2, and E3 proteins. The stability of USP18 is mediated by non-covalent binding to ISG15. IL-6, interleukin-6; TNF, tumor necrosis factor; RT, reverse transcription; dNDPs, deoxyribonucleoside diphosphates; dNTPs, deoxynucleotide triphosphates.

**Figure 2 viruses-16-00485-f002:**
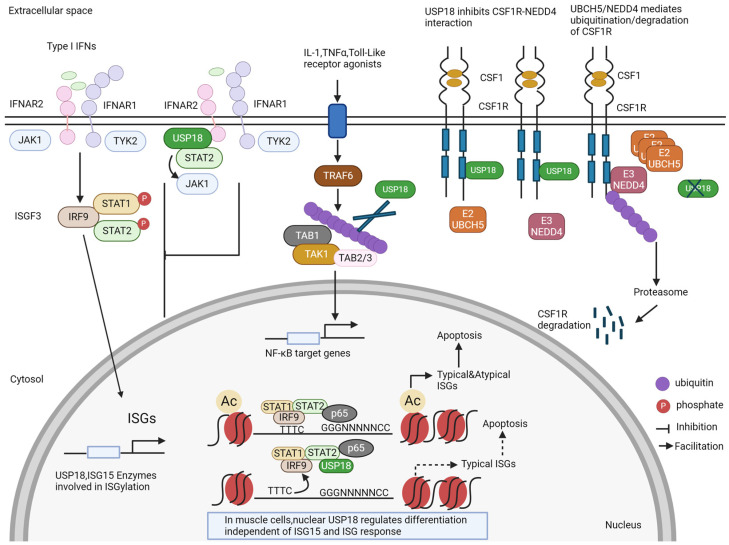
ISG15-independent functions of USP18. In addition to the removal of ISG15 from ISGylated proteins (see Figure 1), USP18 has several other functions. It blocks the signaling of the interferon receptor by competing with JAK1. For some proteins, USP18 can act as an enzyme that removes ubiquitin from ubiquitinated proteins, e.g., for the TAB1–TAK1 complex. USP18 can also bind to the colony-stimulating factor receptor (CSFR) and stabilize it by preventing its degradation. New findings show that USP18 is also a nuclear protein. In the nucleus, USP18 is present in a complex with other transcription factors and thereby regulates the expression of ISGs and cell death. In muscle cells, nuclear USP18 was found to regulate cell differentiation in an ill-defined mechanism.

**Figure 3 viruses-16-00485-f003:**
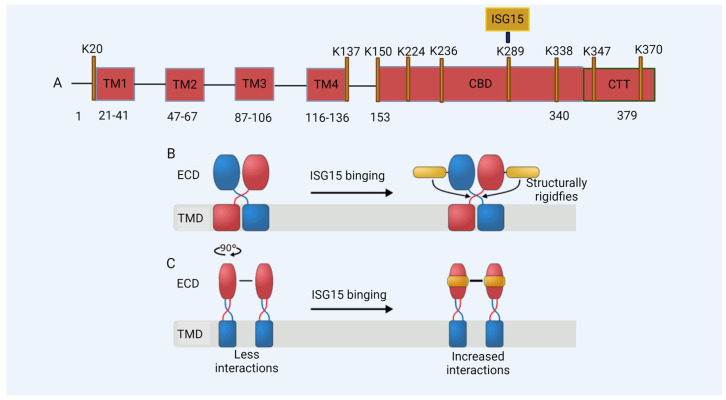
De-ISGylated STING fails to induce interferons. The ISGylation of K289 regulates the activation of STING in DNA sensing. (**A**) A model of the STING protein; lysine (K) residues are shown. TM: transmembrane domain; CBD: cyclic guanosine monophosphate–adenosine monophosphate (cGAMP) binding domain; CTT: C-terminal domain. (**B**,**C**) A schematic of the effect of ISG15 on STING. (**B**) The binding of ISG15 (gold) to the STING dimer (subunits colored in red and blue, respectively) structurally rigidifies the region involved in the rotation of STING during activation, located between the transmembrane (TMD) and the extracellular domain (ECD). (**C**) The binding of ISG15 can increase the number of interactions at the STING dimer-of-dimers interface and thus might stabilize STING oligomerization.

**Figure 4 viruses-16-00485-f004:**
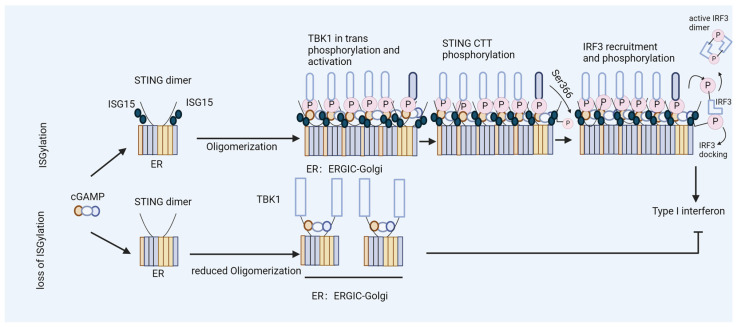
ISGylation of STING fosters efficient type I interferon induction. Model of ISGylated STING, which efficiently oligomerizes after cGAMP binding and triggers type I interferon production. In contrast, STING that is not ISGylated at K289 fails to form oligomers during the activation step and induces limited levels of interferons.

**Figure 5 viruses-16-00485-f005:**
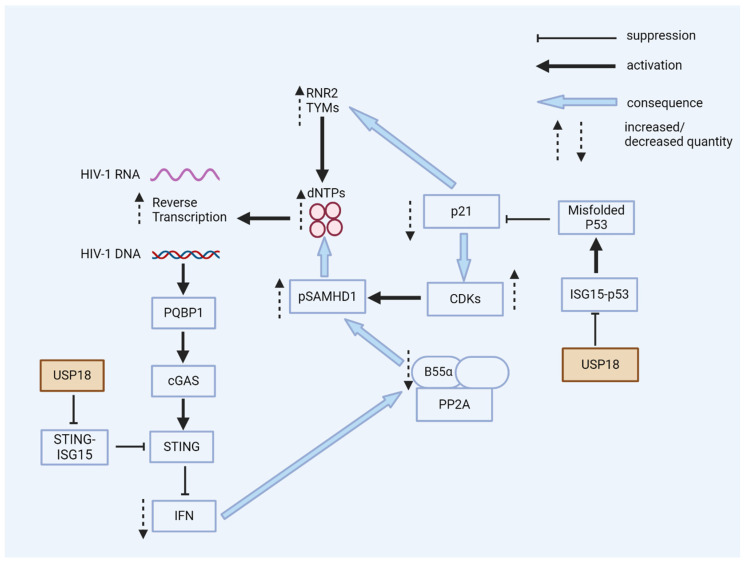
USP18 enhances HIV-1 replication. USP18 has multiple pro-viral actions in the early stages of HIV-1 infection. USP18 expression causes the accumulation of dominant-negative misfolded p53. Without the expression of USP18, misfolded p53 is marked by ISG15 and then degraded. Dominant-negative p53 reduces the expression of p21; as a consequence, CDKs are expressed and phosphorylate the HIV-1 restriction factor SAMHD1, a dNTPase, thereby blocking the antiviral activity of SAMHD1. Low p21 expression causes the expression of enzymes for the de novo synthesis of dNTPs. Higher cellular levels of dNTPs enhance HIV-1’s reverse transcription, especially in resting myeloid cells. The PQBP1/cGAS/STING pathway can sense the viral DNA of HIV-1, inducing interferons and ISGs. However, the expression of USP18 causes the de-ISGylation of STING. De-ISGylated STING fails to oligomerize, with no induction of interferons (see Figure 3 and Figure 4). Low levels of interferons are, in general, beneficial for HIV-1’s replication; antiviral ISGs are not induced and the expression of B55α is not enhanced. B55α is a regulatory subunit of the PP2A phosphatase that eliminates the phosphorylation of SAMHD1 to achieve antiviral activity.

## Data Availability

No new data were created.

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
