# Peer review of "The ISG15-Protease USP18 Is a Pleiotropic Enhancer of HIV-1 Replication"

_viruses, 2024, doi:10.3390/v16040485_

Round 1

Reviewer 1 Report

Comments and Suggestions for Authors

In the manuscript titled “The ISG15-protease USP18 is a pleiotropic enhancer of HIV-1 infection” Lin et al. highlight the literature related to work that they recently published. The title of this work suggests from the beginning that this review is not exploring a field of study as much as focusing on a very specific finding. While that may be OK, this manuscript could very much benefit from a roadmap at the beginning that tells the reader from the outset why they are learning about the various elements of the system and how they will eventually fit together.

Other points:

Line 22 the term “reversed” should be reconsidered. It seems that it’s blocked. Reversed suggests perhaps degradation of interferon itself.

Line 68: “HIV utilizes its viral core to protect its reverse transcripts from detection by cGAS [12].” Seems to be in direct contradiction to the next line that describes cGAS being recruited to the core. If this is a point of contention it should be expressed as such. For example, “while capsid is thought to shield… evidence also exists that capsid attracts…”

The figures are inconsistent. Figure 2 has a legend for the symbols while figure 1 does not. In Figure 1 Particle is spelled Partical and uncoating is symbolized two different ways (cytoplasm vs nuclear). While SAMHD1 is shown to block RT, it is shown to do so downstream of the nucleotide pool, when more accurately it does deplete the dNTP pool. Finally, at the right-hand side of the diagram, viral RNA is shown without an apparent origin. Is this newly transcribed or from incoming virions.

Line 116: has an orphaned reference to a reference (REF).

Line 134-136: “Although ISG15 is deficient in a signal peptide for secretion, it has been detected in the serum of type I IFN-treated patients and in virally infected mice [45, 46].” May be more clear as “Although ISG15 lacks a signal peptide for secretion, it has been detected in the serum of type I IFN-treated patients and in virally infected mice [45, 46].” This is a subtle point, but deficient implies that it would otherwise have a signal when its sequence shows that it was never intended to have one.

Line 153: “Another property of ISG15 is the induction of E-cadherin expression on human dendritic cells, which possibly influences their migratory behavior [54].” Please consider rewording to “ISG15, by inducing E-cadherin expression in human dendritic cells, may impact the migration of these cells.”

Line 156: “ISG15-included exosomes and microparticles contribute to stimulate macrophages to regulate the transmission of anti-HIV activity and release proinflammatory cytokines, respectively [48].” It’s not clear what this means. Do the exosomes and microparticles carry ISG15?

Line 165 appears to be missing a period.

Line 169 it’s not clear what “ubiquitin-prone aggregates” are. Aggregate-prone proteins are described elsewhere, but “ubiquitin-prone aggregates” are mentioned only in the paper referenced. This should be clarified.

Line 201: should this read “ubiquitinated” rather than “ubiquitin-targeted”?

Line 214: should read “cancer cells”, plural.

Line 233: This sentence is very confusing and should be reworded.

Line 246, very minor but 2 periods.

Line 248: “enzymatic-independent” should be replaced for clarity.

The sentence starting on line 269 needs more commas…

Line 296 does not actually describe the mutations. As mutation implies, it’s a change from R to what?

Line 324-: this sentence is not clear. SAMHD1 restriction to HIV-1 infection. It should probably be broken into two sentences.

Line 355: SAMHD1 has many functions that are associated with its phosphorylation state. Some of these are activated by phosphorylation. Here the anti-HIV-1 function is activated by dephosphorylation.

Line 356: “…non-cycling monocyte-derived macrophages (MDMs) are also controlled by PP2A, that renders SAMHD1 antivirally active through de-phosphorylation [132].” Are the cells or SAMHD1 controlled by PP2A? or is SAMHD1 anti-HIV-1 activity in these cells controlled by PP2A?

Line 357: The term “dynamic” is not clear here.

Line 373: This sentence is not clear for several reasons. The presence or absence of Vpx should have the reader thinking “in the presence or absence of SAMHD1.” It would be clearer to say that and then indicate how the presence or absence of SAMHD1 was achieved. Further, it took a while to decipher that “THP-1. USP18” was not the junction of two sentences but rather the apparent designation of a cell line! This is NOT a common cell line and the reader would need to refer to a reference not in this sentence to discover what these cells are. This should obviously be indicated here.

Sentences starting on lines 376 and 378 should include references

Line 380 has the “THP-1. USP18” problem again. The space should be removed.

Line 433 should probably read “studies.” If indeed studies, should there be more than one reference?

Line 437 should be interferon responses

Figure 4 is confusing.

Comments on the Quality of English Language

The English is fine for the most part. The issues are more with the precision of the statements and proofreading.

Author Response

Response to REVIEWER 1

  • We like to thank the reviewer for his/her careful comments!

In the manuscript titled “The ISG15-protease USP18 is a pleiotropic enhancer of HIV-1 infection” Lin et al. highlight the literature related to work that they recently published. The title of this work suggests from the beginning that this review is not exploring a field of study as much as focusing on a very specific finding. While that may be OK, this manuscript could very much benefit from a roadmap at the beginning that tells the reader from the outset why they are learning about the various elements of the system and how they will eventually fit together.

 We agree and included the needed road signs in the first chapter.

Other points:

Line 22 the term “reversed” should be reconsidered. It seems that it’s blocked. Reversed suggests perhaps degradation of interferon itself.

 We agree, we modified the text.

Line 68: “HIV utilizes its viral core to protect its reverse transcripts from detection by cGAS [12].” Seems to be in direct contradiction to the next line that describes cGAS being recruited to the core. If this is a point of contention it should be expressed as such. For example, “while capsid is thought to shield… evidence also exists that capsid attracts…”

We agree, we included more explanations and references to this debate.

The figures are inconsistent. Figure 2 has a legend for the symbols while figure 1 does not. In Figure 1 Particle is spelled Partical and uncoating is symbolized two different ways (cytoplasm vs nuclear). While SAMHD1 is shown to block RT, it is shown to do so downstream of the nucleotide pool, when more accurately it does deplete the dNTP pool. Finally, at the right-hand side of the diagram, viral RNA is shown without an apparent origin. Is this newly transcribed or from incoming virions

We agree, we made several changes in Fig. 1.

Line 116: has an orphaned reference to a reference (REF).

Issue was fixed.

Line 134-136: “Although ISG15 is deficient in a signal peptide for secretion, it has been detected in the serum of type I IFN-treated patients and in virally infected mice [45, 46].” May be more clear as “Although ISG15 lacks a signal peptide for secretion, it has been detected in the serum of type I IFN-treated patients and in virally infected mice [45, 46].” This is a subtle point, but deficient implies that it would otherwise have a signal when its sequence shows that it was never intended to have one.

We agree, we modified the text.

Line 153: “Another property of ISG15 is the induction of E-cadherin expression on human dendritic cells, which possibly influences their migratory behavior [54].” Please consider rewording to “ISG15, by inducing E-cadherin expression in human dendritic cells, may impact the migration of these cells.”

We agree, we modified the text.

Line 156: “ISG15-included exosomes and microparticles contribute to stimulate macrophages to regulate the transmission of anti-HIV activity and release proinflammatory cytokines, respectively [48].” It’s not clear what this means. Do the exosomes and microparticles carry ISG15?

We agree, we modified the text.

Line 165 appears to be missing a period.

Issue was fixed.

Line 169 it’s not clear what “ubiquitin-prone aggregates” are. Aggregate-prone proteins are described elsewhere, but “ubiquitin-prone aggregates” are mentioned only in the paper referenced. This should be clarified.

 We agree, a change in words was done: “promotes the autophagic clearance of ISG15 conjugates”

Line 201: should this read “ubiquitinated” rather than “ubiquitin-targeted”?

Issue was fixed.

Line 214: should read “cancer cells”, plural. 

Issue was fixed.

Line 233: This sentence is very confusing and should be reworded

We agree, we modified the text.

Line 246, very minor but 2 periods.

Issue was fixed.

Line 248: “enzymatic-independent” should be replaced for clarity

Issue was fixed.

The sentence starting on line 269 needs more commas

Issue was fixed.

Line 296 does not actually describe the mutations. As mutation implies, it’s a change from R to what? 

Issue was fixed.

Line 324-: this sentence is not clear. SAMHD1 restriction to HIV-1 infection. It should probably be broken into two sentences.

We agree, we modified the text.

Line 355: SAMHD1 has many functions that are associated with its phosphorylation state. Some of these are activated by phosphorylation. Here the anti-HIV-1 function is activated by dephosphorylation.

We agree, we modified the text.

Line 356: “…non-cycling monocyte-derived macrophages (MDMs) are also controlled by PP2A, that renders SAMHD1 antivirally active through de-phosphorylation [132].” Are the cells or SAMHD1 controlled by PP2A? or is SAMHD1 anti-HIV-1 activity in these cells controlled by PP2A? …

We agree, we modified the text.

Line 357: The term “dynamic” is not clear here.

We agree, we modified the text.

Line 373: This sentence is not clear for several reasons. The presence or absence of Vpx should have the reader thinking “in the presence or absence of SAMHD1.” It would be clearer to say that and then indicate how the presence or absence of SAMHD1 was achieved. Further, it took a while to decipher that “THP-1. USP18” was not the junction of two sentences but rather the apparent designation of a cell line! This is NOT a common cell line and the reader would need to refer to a reference not in this sentence to discover what these cells are. This should obviously be indicated here.

We agree, we modified the text.

Sentences starting on lines 376 and 378 should include references

Issue was fixed.

Line 380 has the “THP-1. USP18” problem again. The space should be removed. …

Issue was fixed.

Line 433 should probably read “studies.” If indeed studies, should there be more than one reference? 

Issue was fixed.

Line 437 should be interferon responses

 Issue was fixed.

Figure 4 is confusing.

We agree, we made several changes in Fig. 4 (now Fig. 5).

Comments on the Quality of English Language

The English is fine for the most part. The issues are more with the precision of the statements and proofreading.

Reviewer 2 Report

Comments and Suggestions for Authors

The authors review the roles of ISG15, USP18, STING, UBPQ1, IFN, SAMHD1 and other host factors in the regulating the replication of HIV.

The authors have written a thorough and scholarly review of their views of this complex regulatory network.  However, while the article is impressive in its scope, it is difficult to read.  The figures and text are so complex that few readers will be able to follow them.  The review needs to be re-written with the goal of limiting its scope and making the text easier to follow.  Minor corrections will not suffice.  The review has to enlighten readers without the need to expend extensive time and effort. 

The authors present their viewpoint, which they are free to do, but consideration of the work of others and a realization that there are other viewpoints should be conveyed.  The question of HIV nucleic acid sensing is viewed differently by different researchers and consensus on many of the points described here has not been reach.  The authors present it as though they were established facts, which is not the case.  Sensing of HIV DNA by cGAS has been disputed by several groups.  To name 3 out of many, Geoffinet’s group and Towers’ group both show lack of cGAS sensing (EMBO J. 2020 Oct 15; 39(20): e103958, https://doi.org/10.1073/pnas.2002481117) as does Goff’s lab. 

The text has grammatical mistakes and run-on sentences.  Revise throughout. 

Clarify throughout when referring to macrophages or T cells.  In general, it refers to the roles of ISG15, USP18 and PQBP1 without stating the cell-type, making it sound like the principles hold for T cells and macrophages, yet these were mainly described in macrophages and may not pertain in T cells. 

The Abstract is complex, confusing and contains grammatical mistakes, a few of which are detailed here:

  “IFN production can be reversed” does not make sense.  Perhaps “suppressed”?

Line 25.  The word USP18 is repeated in one sentence.  Delete “and USP18” and substitute “which”

Line 29.  Insert hyphen into cGAS-STING mediated.  cGAS-STING-mediated

Line 32.  “ISGylation by USP18” appears to be a mistake.  This should be by USG15.

Line 33.  This is unrelated to the topic at hand and only serves only as a distraction.

Lines 24-25.  This is a run-on sentence.

Line 69.  “The cGAS protein”.  Delete “The” and “protein”

Figure 1.  The figure has so many arrows and pathways that it is incomprehensible.  

The legend mentions HIV in endosomes but no viral RNA is shown in the endosome.  

The figure gives the impression that these pathways are universal but they are not.  They are mainly macrophage-specific. 

“Partical” is misspelled. 

NFkb doesn’t seem to do anything. 

The circular pathway on the right does not seem to do anything. 

Rig-I is shown as sensing viral RNA but that is not accepted. 

Fig. 2.  The figure shows multiple unrelated functions of USP18.  At least 5 functions, including binding the CSF-R, binding IFNAR, acting in the nucleus to bind transcription factors, etc.  And these are in addition to its role as a protease that removes ISG15.  It seems unlikely that a single protein has so many completely unrelated functions. 

Line 367.  Remove “is enhancing” replace with "enhances".

Line 469.  “showed” not “identified”

Comments on the Quality of English Language

Review throughout for grammar and clarity.  break run-on sentences (those with multiple phrases) into 2 or 3 sentences. 

Author Response

Response to REVIEWER 2

  • We like to thank the reviewer for his/her careful comments!

The authors review the roles of ISG15, USP18, STING, UBPQ1, IFN, SAMHD1 and other host factors in the regulating the replication of HIV.

The authors have written a thorough and scholarly review of their views of this complex regulatory network.  However, while the article is impressive in its scope, it is difficult to read.  The figures and text are so complex that few readers will be able to follow them.  The review needs to be re-written with the goal of limiting its scope and making the text easier to follow.  Minor corrections will not suffice.  The review has to enlighten readers without the need to expend extensive time and effort. 

The authors present their viewpoint, which they are free to do, but consideration of the work of others and a realization that there are other viewpoints should be conveyed.  The question of HIV nucleic acid sensing is viewed differently by different researchers and consensus on many of the points described here has not been reach.  The authors present it as though they were established facts, which is not the case.  Sensing of HIV DNA by cGAS has been disputed by several groups.  To name 3 out of many, Geoffinet’s group and Towers’ group both show lack of cGAS sensing (EMBO J. 2020 Oct 15; 39(20): e103958, https://doi.org/10.1073/pnas.2002481117) as does Goff’s lab. 

We agree, all these topics are addressed now.

The text has grammatical mistakes and run-on sentences.  Revise throughout. 

We agree and fixed hopefully all issues.

Clarify throughout when referring to macrophages or T cells.  In general, it refers to the roles of ISG15, USP18 and PQBP1 without stating the cell-type, making it sound like the principles hold for T cells and macrophages, yet these were mainly described in macrophages and may not pertain in T cells. 

We mentioned this clearly at several places, as in Fig. 1 legends.

The Abstract is complex, confusing and contains grammatical mistakes, a few of which are detailed here:

  “IFN production can be reversed” does not make sense.  Perhaps “suppressed”?

--> Issue was fixed

Line 25.  The word USP18 is repeated in one sentence.  Delete “and USP18” and substitute “which”

--> Issue was fixed

Line 29.  Insert hyphen into cGAS-STING mediated.  cGAS-STING-mediated

--> Issue was fixed

Line 32.  “ISGylation by USP18” appears to be a mistake.  This should be by USG15.

--> Issue was fixed

Line 33.  This is unrelated to the topic at hand and only serves only as a distraction.

 --> We deleted this sentence.

Lines 24-25.  This is a run-on sentence

--> We do not get this point.

Line 69.  “The cGAS protein”.  Delete “The” and “protein”

--> Issue was fixed

Figure 1.  The figure has so many arrows and pathways that it is incomprehensible.  

The legend mentions HIV in endosomes but no viral RNA is shown in the endosome.  

The figure gives the impression that these pathways are universal but they are not.  They are mainly macrophage-specific. 

“Partical” is misspelled. 

NFkb doesn’t seem to do anything. 

The circular pathway on the right does not seem to do anything. 

Rig-I is shown as sensing viral RNA but that is not accepted. 

We made several changes in Fig. 1 addressing the discussed issues.

Fig. 2.  The figure shows multiple unrelated functions of USP18.  At least 5 functions, including binding the CSF-R, binding IFNAR, acting in the nucleus to bind transcription factors, etc.  And these are in addition to its role as a protease that removes ISG15.  It seems unlikely that a single protein has so many completely unrelated functions. 

--> We are also impressed by the diverse functions of USP18 described in the literature. The future will tell how relevant each described functions is.

Line 367.  Remove “is enhancing” replace with "enhances".

--> Issue was fixed

Line 469.  “showed” not “identified”

 --> Issue was fixed

Comments on the Quality of English Language

Review throughout for grammar and clarity.  break run-on sentences (those with multiple phrases) into 2 or 3 sentences. 

--> Many related issues of long and complex sentences were fixed.

Reviewer 3 Report

Comments and Suggestions for Authors

Lin and colleagues wrote an excellent review on how ISG15 and USP18 regulate HIV replication. The manuscript thoroughly covers the related literature with concise and clear figures and will help us understand the role of these proteins in HIV biology. There are only minor suggestions:

1. It would be nice to guide readers to which figures they should refer, i.e., “see Figure 1” etc.

2. In the first paragraph “ 1.1. Interferon and interferon induction by HIV-1”, it would be helpful to briefly describe HIV RNA sensing pathways as they are depicted in the Figure 1.

Comments on the Quality of English Language

There are some typos and errors e.g., “Similar” in Line 418 (Similarly).    

Author Response

Response to REVIEWER 3

  • We like to thank the reviewer for his/her careful comments!

Lin and colleagues wrote an excellent review on how ISG15 and USP18 regulate HIV replication. The manuscript thoroughly covers the related literature with concise and clear figures and will help us understand the role of these proteins in HIV biology. There are only minor suggestions:

  1. It would be nice to guide readers to which figures they should refer, i.e., “see Figure 1” etc

We agree and included the missing figure references.

  1. In the first paragraph “ 1.1. Interferon and interferon induction by HIV-1”, it would be helpful to briefly describe HIV RNA sensing pathways as they are depicted in the Figure 1.

We agree and included information about the current models for HIV RNA sensing.

Comments on the Quality of English Language

There are some typos and errors e.g., “Similar” in Line 418 (Similarly).   …

We fixed these isssues throughout the ms. at related positions.

Reviewer 4 Report

Comments and Suggestions for Authors

This is a comprehensive and clearly written review of the role of the ISG15-protease USP18 in enhancing HIV-1 replication.  The Authors do a good job of summarizing this area of research and highlighting their own contributions.  This Reviewer has no concerns with the manuscript and feels that it will be useful to the field.  Some minor editing for the Authors' consideration are:

1.  The title states that USP18 is an enhancer of HIV-1 "infection," but it really is an enhancer of "replication" as it plays no role in entry of the virion into the cell.

2.  Lines 101-102.  These lines cite two doctoral theses in all upper case letters.  This may be a formatting mistake and is quite confusing.

3.  Line 318 makes the claim that p53 "perfectly" inhibits the IL-6 promoter.  p53 may be doing this effectively, but to claim perfection is a stretch.

4.  Lines from 372- 390.  Many of the THP-1 engineered cell line names contain a space between THP1 and the cell identity.  This is quite confusing as it look reads as if the sentence ends after THP1.  The space should be removed

Author Response

Response to REVIEWER 4

  • We like to thank the reviewer for his/her careful comments!

Comments and Suggestions for Authors

This is a comprehensive and clearly written review of the role of the ISG15-protease USP18 in enhancing HIV-1 replication.  The Authors do a good job of summarizing this area of research and highlighting their own contributions.  This Reviewer has no concerns with the manuscript and feels that it will be useful to the field.  Some minor editing for the Authors' consideration are:

  1. The title states that USP18 is an enhancer of HIV-1 "infection," but it really is an enhancer of "replication" as it plays no role in entry of the virion into the cell. …

We agree, we made such changes in title and throughout the ms. at related positions

  1. Lines 101-102.  These lines cite two doctoral theses in all upper case letters.  This may be a formatting mistake and is quite confusing. …

We fixed this issue.

  1. Line 318 makes the claim that p53 "perfectly" inhibits the IL-6 promoter.  p53 may be doing this effectively, but to claim perfection is a stretch.

We agree, we changed the text accordingly.

  1. Lines from 372- 390.  Many of the THP-1 engineered cell line names contain a space between THP1 and the cell identity.  This is quite confusing as it look reads as if the sentence ends after THP1.  The space should be removed …

We fixed these issues.

Round 2

Reviewer 2 Report

Comments and Suggestions for Authors

The initial submission was difficult to read, with many grammatical errors. It was termed complex and scattered, covering too many topics and the figures were confusing.  The authors have made many minor corrections and modifications to the text and fixed the errors pointed out by the reviewer.  This has helped but the review still contains a lot of grammatical errors and remains scattered.  The majority of the revisions that have been made contain grammatical errors.  The text still needs a thorough review for grammar and clarity.      

Comments on the Quality of English Language

The authors have made many minor corrections and modifications to the text and fixed the errors pointed out by the reviewer.  This has helped but the review still contains a lot of grammatical errors and remains scattered.  The majority of the revisions that have been made contain grammatical errors.  The text still needs a thorough review for grammar and clarity.  

Author Response

The new version has several text corrections.